# Novel NPR2 Gene Mutations Affect Chondrocytes Function via ER Stress in Short Stature

**DOI:** 10.3390/cells11081265

**Published:** 2022-04-08

**Authors:** Qiuyue Li, Xin Fan, Wei Lu, Chengjun Sun, Zhou Pei, Miaoying Zhang, Jinwen Ni, Jing Wu, Fa-Xing Yu, Feihong Luo

**Affiliations:** 1Department of Pediatric Endocrinology and Inherited Metabolic Diseases, Children's Hospital of Fudan University, National Children's Medical Center, Shanghai 201102, China; 17211290018@fudan.edu.cn (Q.L.); wei_lu@fudan.edu.cn (W.L.); 0356117@fudan.edu.cn (C.S.); zpei09@fudan.edu.cn (Z.P.); 17211240009@fudan.edu.cn (M.Z.); 16111240022@fudan.edu.cn (J.N.); 13111240013@fudan.edu.cn (J.W.); 2Department of Pediatrics, The Second Affiliated Hospital of Guangxi Medical University, Nanning 530003, China; fanxin602@163.com; 3Institute of Pediatrics, Children’s Hospital of Fudan University, National Children's Medical Center, Shanghai 201102, China; 4Shanghai Key Laboratory of Medical Epigenetics, Institutes of Biomedical Sciences, Shanghai Medical College, Fudan University, Shanghai 200032, China

**Keywords:** chondrocyte apoptosis, ER stress, NPR2, N-glycosylation, short stature

## Abstract

Natriuretic peptide receptor 2 (NPR2) plays a key role in cartilage and bone morphogenesis. The *NPR2* gene mutations result in acromesomelic dysplasia, Maroteaux type (AMDM), short stature with nonspecific skeletal abnormalities (SNSK), and epiphyseal chondrodysplasia, Miura type (ECDM). However, the pathogenic mechanism remains unclear. In our study, we identified one de novo (R557C) and six novel variants (G602W, V970F, R767*, R363*, F857S, and Y306S) in five independent Chinese families with familial short stature. Three patients with heterozygous mutations (G602W, V970F, and R767*) were diagnosed with SNSK (height SD score ranged from −2.25 to −5.60), while another two with compound heterozygous mutations (R363* and F857S, R557C and Y306S) were diagnosed with AMDM (height SD score ranged from −3.10 to −5.35). Among three patients with heterozygous status, two patients before puberty initiation with rhGH treatment significantly improved their growth (height velocity 7.2 cm/year, 6.0 cm/year), and one patient in puberty had a poor response to the rhGH treatment (height velocity 2.5 cm/year). Seven *NPR2* gene variants were constructed and overexpressed in HEK293T and ATDC5 cells, and we found that ATDC5 cells with mutant *NPR2* gene showed decreased differentiation, as evidenced by lower expression of *ColII*, *ColX*, and *BMP4* and higher expression of *Sox9*. Moreover, the apoptosis rate was elevated in ATDC5 cells expressing the mutant *NPR2* gene. N-glycosylation modification, plasma membrane localization, and ER stress resulted from the accumulation of mutant protein in ER, as shown by the higher expression of GRP78 and p-IRE1α. Overall, our results provide a novel insight into NPR2 loss of function, which could promote chondrocyte apoptosis and repress cell differentiation through ER stress and the unfolded protein response.

## 1. Introduction

Genetic factors play prominent roles in normal human growth. Previous studies showed that *NPR2* gene variants can manifest as isolated short stature or in conjunction with skeletal dysplasia [1]. However, the pathogenic mechanism remains unclear. The *NPR2* gene encodes the type C natriuretic peptide (CNP) receptor, a cell surface plasma membrane protein, which structurally comprises five domains, including CNP-binding, transmembrane, tyrosine kinase, coil–coil, and guanylate cyclase domain [2]. The *NPR2* plays a key role in transducing extracellular signals into cells to influence intracellular cyclic guanosine monophosphate (cGMP) concentrations and controls fundamental cellular processes, including cell proliferation and differentiation [3]. 

*NPR2* is involved in the longitudinal growth of long bones. Loss-of-function (LOF) mutations in *NPR2* cause acromesomelic dysplasia, Maroteaux type (AMDM #602875), and short stature with nonspecific skeletal abnormalities (SNSK #616255) [4,5], while gain-of-function (GOF) mutations have been reported to be responsible for epiphyseal chondrodysplasia, Miura type (ECDM #615923) [6]. Currently, two pathogenic mechanisms have been proposed for *NPR2* gene alterations, that is, disruption of NPR2 trafficking to the plasma membrane and subsequent CNP binding process and inhibition of its own catalytic activity by conformational changes in NPR2 mutants [7,8]. However, the pathogenic mechanisms and the functions of the altered *NPR2* gene are not fully understood in chondrocytes yet. 

Before plasma membrane localization, NPR2 undergoes a number of protein maturation steps, such as folding, assembly, post-translational modification, and sorting in the ER [9]. Normally, poorly folded proteins were confirmed to be retained in ER and targeted for degradation, and this ER protein quality control mechanism may be overwhelmed by persistent insults, such as accumulation of mutant proteins, resulting in ER stress (ERS) [10]. It has been reported that the unfolded protein response (UPR) could be activated while ERS occurs through an adaptation mechanism. In that case, the UPR attenuates protein folding via activation of transcription factor 6 and induces ER-associated protein degradation to eliminate misfolded proteins via inositol-requiring enzyme 1 (IRE1) signaling [11]. However, cellular apoptosis can also be activated when stress exceeds the capacity of the ER homeostatic machinery. Recent studies suggest that ERS caused by misfolded mutant proteins contributes to the molecular pathology in several skeletal dysplasia disorders [12,13]. Thus, we speculated that the function of the ERS–UPR-apoptosis signaling axis was evoked by mutated *NPR2* genes in chondrocytes. 

In this study, we identified six novel *NPR2* gene mutations and one de novo *NPR2* gene mutation in five families with short stature. We further investigated their function, expression, and subcellular abnormal localization of the mutant NPR2 protein and studied the differentiation and apoptosis of chondrocytes to explore the potential pathogenesis of *NPR2* gene mutation resulting in skeletal dysplasia.

## 2. Material and Methods

### 2.1. Pedigree Analysis of Children with Short Stature

Patients were recruited from the Department of Pediatric Endocrinology and Inherited Metabolic Diseases, Children’s Hospital of Fudan University. WES was performed on five independent Chinese families with familial short stature fulfilling the following diagnostic criteria: height standard deviation score (HSDS) ≤ −2, excluding chromosome abnormality, diagnosed congenital bone diseases, and secondary short stature. Birth weights, birth body lengths, and heights at each visit were accessed. Small for gestational age (SGA, defined as birth height or birth weight <P3 for gestational age) was defined using the national standards [14]. This study was approved by the Children’s Hospital Ethics Committee of the Children’s Hospital of Fudan University. Written informed consent was obtained from all participants or legal guardians.

### 2.2. Genetic Testing

Whole-exome sequencing (WES) was performed by WuXi NextCODE Genomics, Shanghai, China (CLIA Lab ID: 99D2064856). Briefly, exome capture was performed using the Agilent SureSelect Human All Exon V5, Illumina TruSeq Rapid PE Cluster, and SBS kits (Agilent Technologies, Santa Clara, CA, USA). Illumina HiSeq provided the 2000/2500 platform for WES. Read alignment was referenced to the human genome sequence (http://genome.ucsc.edu, accessed on 7 July 2000) using the Burrows-Wheeler Aligner v.0.6.2. Duplicate paired-end reads were marked with Picard v.1.55 (https://broadinstitute.github.io/picard/, accessed on 5 September 2020). The base quality score recalibration, indel realignment, and variant discovery were performed using the Genome Analysis Toolkit v.2.3–9. Variants were annotated using a pipeline developed in-house and filtered in the Genome Aggregation Database (gnomAD), Exome Variant Server, dbSNP databases, and Exome Aggregation Consortium. All putative pathogenic variants observed by ES were validated by Sanger sequencing (primer sequences and amplification protocols are available upon request). 

### 2.3. Wild-Type and Mutant NPR2 Gene Expression Plasmid Construction 

The cDNA of wild-type and mutant (T2570C and C1669T) *NPR2* gene with a DYKDDDDK-Tag (Flag) at its C-terminus, referred to as WT^Flag^ and MUT^Flag^, respectively, were inserted into the pCDH-CMV-MCS-EF1-copGFP lentiviral expression vector. The remaining mutant (G1804T, G2908T, C2299T, C1087T, T2570C, A917C, C1669T) *NPR2* gene coding sequences were also Flag-tagged at the C-terminus and inserted into the pCDH-CMV-MCS-EF1-Puro lentiviral expression vector.

### 2.4. Lentivirus Preparation 

Lentiviral vectors were produced in HEK293T cells with packaging plasmids. The recombinant viral supernatants were harvested and then used to infect target cells (HEK293T and ATDC5 cell lines) in the presence of 8 μg/mL polybrene. 

### 2.5. Cell Culture and Transfection and Generation of Stable Cell Lines

HEK293T cells were cultured in Dulbecco’s modified Eagle’s medium (DMEM, Invitrogen, Waltham, MA, USA) supplemented with 10% fetal calf serum (Hyclone, Walthman, MA, USA), 2 mM L-glutamine, and 100 U/mL penicillin/streptomycin at 37 °C with 5% CO_2_. ATDC5 cells were incubated in high-glucose DMEM (Gibco, Gibco Life Technologies, Grand Island, NY, USA) with 5% fetal bovine serum (Hyclone, Walthman, MA, USA), 100 IU/mL penicillin, and 100 mg/mL streptomycin (Cat. #V900929, Sigma Aldrich, St Louis, MO, USA). ATDC5 cells were passaged every 2 d. Chondrogenic differentiation was induced in insulin–transferrin–selenium (ITS) (Roche Diagnostics GmbH, Mannheim, Germany) medium containing insulin, according to previous protocols [15]. For NPR2^WT/MU^-expressing lentiviral vector transfection, cells were seeded into 6-well plates for 24 h to reach 70% confluence, and then NPR2^WT/MU^-expressing lentiviral vectors were transfected into the HEK293T cells or ATDC5 at MOI = 3 and MOI = 5, respectively, in the presence of 8 μg/mL polybrene. Cells samples were harvested at 48 h after transfection, and Western blot was performed using the antibody against Flag (Cat. No. R20008, 1:5000, Abmart, Shanghai, China). To generate stable cell lines, after 48 h, (G1804T, G2908T, C2299T, C1087T, A917C) NPR2-expressing stable cell lines were selected using puromycin with the final concentration of 5 ug/mL in 96-well plates, and (T2570C, C1669T, and wild-type) NPR2-expression stable cell lines were selected by observing the expression of GFP with a fluorescence microscope. The cell lines expressing empty lentiviral vectors were defined as vector control. Cells samples were harvested when the transfection efficiency reached more than 90%.

### 2.6. RNA Extraction and Real-Time-Polymerase Chain Reaction (RT-PCR) Analysis 

ATDC5 cells were incubated in 6-well plates and cultured in a differentiation medium for RNA isolation. To verify the effect of NPR2 mutants on chondrocyte differentiation, we examined the expression of four chondrocyte markers, including Col2A and BMP4 (proliferative zone), Sox9 and Col10a1 (hypertrophic zone). Total RNA was prepared using the Qiagen RNeasy Mini Kit (QIAGEN, Germantown, MD, Germany) from ATDC5 cells expressing *NPR2* mutants (for 7 d), which mimics the heterozygous and compound heterozygous mutation status observed in patients. Total RNA was further purified using the RNeasy Micro Kit (QIAGEN) and RNase-Free DNase Set (QIAGEN). RNA was reverse-transcribed into cDNA using PrimeScript RT reagent according to the protocol (TAKARA, Shiga, Japan). Following cDNA synthesis, quantitative PCR (qPCR) was performed according to the manufacturer’s instructions. The relative gene expression data were analyzed using a 2^−ΔΔCT^ method, and the data were normalized to the reference gene *Gapdh* for mRNA expression analysis. Mouse primers were designed using the Basic Local Alignment Search Tool program by the National Center for Biotechnology Information and synthesized by the Generay Biotech Company.

### 2.7. Annexin V/Propidium Iodide (PI) Flow Cytometry Assay

ATDC5 cells (5 × 10^5^) were seeded into a 6-well plate, and chondrocytes were induced with ITS medium for 7 d. Cells were collected following digestion and centrifugation, washed twice with phosphate-buffered saline (PBS), and 0.5 × 10^6^ cells were stained by adding 500 µL 1× Binding Buffer and 5 µL Annexin V fluorescein isothiocyanate (FITC) reagent (Annexin V-FITC Apoptosis Detection Kit; BD Biosciences, San Jose, CA, USA). Following a 15 min incubation in the dark, 5 µL PI was added for another 5 min. Cells were then analyzed using flow cytometry.

### 2.8. Enzyme-Linked Immunosorbent Assay

Stably transfected HEK293T cells were seeded into 6-well plates, and after 24 h, the cells were washed twice with PBS. Then, the cells were incubated with 100 nM CNP (CNP-22; CAYMAN, Ann Arbor, MI, USA) dissolved in serum-free DMEM. After 30 min incubation at 37 °C, 0.1 M HCl was added to stop endogenous phosphodiesterase activity and stabilize cGMP. Cell lysates were collected, and cGMP levels were measured using the cGMP Complete ELISA kit (ENZO Life Sciences, Madison Avenue, NY, USA). Experiments were conducted in triplicate, significance was determined using the Student’s *t*-test, and *p* < 0.05 was considered significant.

### 2.9. Confocal Microscopy

HEK293T cells were transfected with *NPR2^WT^*^-Flag^ and *NPR2^MUT^*^-Flag^ lentivirus. Anti-FLAG mouse monoclonal (1:2000, R20008, Abmart, Shanghai, China) and anti-ER marker calnexin (1:2000 dilution, ab22595, Abcam, Cambridge, UK) were both diluted 1:2000 used as primary antibodies. Anti-mouse IgG (Alexa Fluor 488-conjugated; CST #4412, Danvers, MA, USA) and anti-rabbit IgG (Alexa Fluor 555-conjugated; CST # 52286S) were used as secondary antibodies, respectively. Cells were counterstained with diamino-2-phenylindole (DAPI; Beyotime Biotechnology, Wuhan, China) to visualize the nucleus. Fluorescent images were captured using a Leica DM5500 B fluorescent microscope (Leica Microsystems GmbH, Wetzlar, Germany). Three biological replicates were performed.

### 2.10. Western Blot Analysis

Cells were harvested with trypsin-ethylenediaminetetraacetic acid (0.25%) (Hyclone), lysed in lysis buffer (50 mM Tris, pH 7.5, 150 mM NaCl, and 1% Triton X-100, Thermo Fisher Scientific, Carlsbad, CA, USA) supplemented with Roche complete protease inhibitor cocktail on ice for 1 h, and then centrifuged (12,000× *g*, 10 min, 4 °C) to remove cell debris and insoluble materials. The supernatant was collected as the total cellular protein. Protein concentration was measured using a bicinchoninic acid assay (Beyotime Biotechnology, Wuhan, China). Proteins were separated in a 10% denaturing Tris-glycine gel, and Western blot analysis was performed using the corresponding antibodies, including mouse anti-Flag-tag monoclonal antibody (1:5000 dilution, Protein Tech, Wuhan, China). The anti-IRE1α (1:1000 dilution, NB100-2324, Novus, Briarwood, NY, USA), anti-IRE1α-p (1:1000 dilution, NB100-2323, Novus), and anti-GRP78 (1:1000 dilution, NBP1-06274, Novus) were purchased from NOVUS; anti-Caspase 3 (1:1000 dilution, 66470-2-Ig, Proteintech, Wuhan, China) and anti-cleave caspase 3 (1:1000 dilution, MA9132, Abmart) were purchased from Proteintech and Abmart company, respectively. GAPDH (1:5000 dilution, # 2118S, CST, Danvers, MA, USA) was purchased from Cell Signaling Technology. Protein band intensity was quantified using Image J software (Image J (nih.gov), accessed on 15 April 2021).

### 2.11. Post-Translational Modification Analysis

Mature membrane proteins undergo complex post-translational modifications, such as glycosylation (N-glycosylation, O-glycosylation), ubiquitination, and phosphorylation. These modifications are often reversed by different enzymes, such as O-glycosidase, Endoglycosidase-H, PYR-41, and Lambda phosphatase [16]. To assess the post-translational status of the NPR2 protein, ATDC5 cells expressing *NPR2^WT^*^-Flag^ or *NPR2^MUT-Flag^* were subjected to chondrogenic differentiation. Protein lysates were treated with endoglycosidases endo-β-N-acetylgucosaminidase H (Endo-H) (750 units, New England Biolabs, Mississauga, ON, Canada), PYR-41 (a cell-permeable irreversible inhibitor of ubiquitin-activating enzyme E1) (MCE, HY-13296, China), or Lambda phosphatase (400,000 units/mL, New England Biolabs), according to the manufacturer recommendations.

### 2.12. Statistical Analysis

Data were analyzed using SPSS software (IBM SPSS Statistics 20, Chicago, IL, USA). Data were expressed as the mean ± standard error (SE). Unless otherwise indicated, data of three independent experiments were analyzed using a one-way ANOVA analysis of variance, and *p* < 0.05 was considered significant.

## 3. Results 

### 3.1. Clinical and Genetic Analyses

The major characteristics of five patients with heterozygous or compound heterozygous *NPR2* gene variants are summarized in Table 1 and Figure 1. Among the five probands, three children were born SGA (proband 3–5). Three patients (proband 1–3) carrying heterozygous *NPR2* gene variants had an HSDS of −2.25 to −5.60, and their affected parent HSDSs were from −4.3 to −3.2. Radiographs showed nonspecific changes in limbs. Two patients (proband 4–5) with compound heterozygous *NPR2* gene variants presented short limbs, their HSDS ranged from −3.10 to −5.35, and their affected parent HSDSs ranged from −2.9 to −0.57. Radiographs showed severely compressed and deformed vertebrae in patient 4 and shortened bilateral iliac bones, irregular acetabular rim, as well as a widened backbone and metaphysis in patient 5 (Appendix A). 

Three patients (proband 1–3) received recombinant human growth hormone (rhGH) therapy, and their height velocity (HV) was 2.5 cm/year, 7.2 cm/year, and 6.0 cm/year at the latest evaluation, respectively (Figure 1C). Patients 4 and 5 did not receive rhGH therapy due to their young age.

### 3.2. Mutant NPR2 Gene Inhibited Chondrocyte Differentiation and Promoted Apoptosis in ATDC5 Chondrocytes

The underlying mechanism of our seven short stature patients carrying mutations in *NPR2* gene could result from chondrocyte slow proliferation, reduced differentiation, or untimely death. Previous studies have established that NPR2 protein increases chondrocytes proliferation by elevating cGMP and activating the cGMP-dependent protein kinase PRKG2 [17]. To understand the expression and function of *NPR2* gene mutants expressed in ATDC5, chondrocytes and their effect on cellular differentiation and apoptosis were evaluated. Western blot analysis revealed that the expression levels of NRP2 mutants (G602W/WT, V970F/WT, R767*/WT, F857S/R363*, and Y306S/R557C) were increased when compared to the wild type. Collagen type II (*COLII*), type X collagen (*COLX*), bone morphogenetic protein 4 (*BPM4*), and SRY-box transcription factor 9 (*SOX9*) were used as markers for chondrocyte differentiation and their expression. As shown in Figure 2B–E, compared to those of wild-type *NPR2*-expressing cells, *COLII*, *COLX*, and *BPM4* expression was significantly downregulated, whereas *SOX9* expression was upregulated in cells expressing different *NPR2* mutants (G602W/WT, V970F/WT, R767*/WT, F857S/R363*, and Y306S/R557C). Hence, these *NPR2* gene mutants inhibited chondrocytes differentiation.

Previously, investigators studied that mutated collagen was retained in the ER and caused ERS, activating UPR signaling and causing apoptosis [18]. In our cellular model, confocal microscope analysis showed that the mutant NPR2 proteins were retained intracellularly. We assessed the function of *NPR2* gene mutants on cell apoptosis, cellular Annexin V, and PI levels, and the protein level of cleaved caspase-3 were analyzed. As shown in Figure 2F,H, *NPR2* gene mutant-expressing ATDC5 cells exhibit elevated apoptosis. The protein levels of cleaved caspase-3 were increased in *NPR2* gene mutant-expressing ATDC5 cells compared to wild type. Nevertheless, cleaved caspase-3 was similar in NPR2 wild type and vector control (Figure 2G,I). Together, these data indicated that *NPR2* gene variants promoted chondrocytes apoptosis. 

### 3.3. Molecular Characterization of NPR2 Gene Variants

The competency of *NPR2* mutants in mediating CNP-induced signaling was assessed. *NPR2^MUT^* led to a significant decrease in CNP-induced cGMP production compared to that of *NPR2^WT^*. To mimic the heterozygous or compound heterozygous state of these patients in vitro, G602W, V970F, and R767* were co-expressed with wild type, F857S was co-expressed with R363*, and Y306S was co-expressed with R557C in HEK293T cells, and it was noticed that all the combinations led to severe loss in the CNP-induced cGMP response compared to that of *NPR2^WT^* (Figure 3).

### 3.4. The Expression of Mutant and Wild-Type NPR2 Gene in HEK293T Cell Lines

To assess the effect of each NPR2 gene variant, NPR2^WT^ and NPR2^MUT^ were expressed in HEK293T cells. R767* and R363* exhibited a smaller molecular weight than NPR2^WT^, consistent with truncated mutations. The molecular weights of the remaining mutants were similar to NPR2^WT^ (Figure 4A). Western blot analysis also revealed that the expression levels of NRP2 F857S and NPR2 R557C were reduced when compared to the wild type, while the remaining six mutants were similarly expressed as wild type (Figure 4B). 

### 3.5. Subcellular Localization of NPR2^WT^ and NPR2^MUT^ Proteins

As shown by confocal microscopy, the *NPR2* mutants (except R557C) identified in this study were not present in the plasma membrane when expressed in HEK293T cells (Figure 5). By labeling the ER with calnexin protein, it has been shown that *NPR2* mutants (except R557C) co-localize with the calnexin protein. These results indicated that six *NPR2* mutants (G602W, V970F, R767*, R363*, F857S, and Y306S) failed to be transported out of the ER. Hence, plasma membrane targeting was blocked (Figure 5). In addition, the cell imaging of ER revealed morphological changes in HEK293T cells expression. In wild-type cells, ER exhibited the homogeneous and loose structure distributed in the cytoplasm. However, ER in the NPR2^MUT^ group aggregated near the nuclear envelope (Figure 5).

### 3.6. NPR2 Gene Mutants Activated the ERS-UPR in ATDC5 Chondrocytes

The *NPR2^WT^* protein, due to post-translational modification, was detected in Western blotting as two bands with distinct molecular weights. The upper band was down-shifted following EndoH digestion, indicating that the upper band was N-glycosylated, while the lower band represented proteins without N-glycosylation (Figure 6A). Except for R557C, all *NPR2* mutants exhibited as a single band in Western blotting, indicating that these mutants were not N-glycosylated. Moreover, when the R557C mutant was co-expressed with Y306S, its N-glycosylation was also repressed (Figure 6B).

N-glycosylation is important for protein folding and sorting in the ER, and *NPR2* mutant defective in N-glycosylation may not be properly folded and induce the UPR and ERS. The cellular “quality control system” recognizes misfolded proteins, which leads to two outcomes: (1) ER-related degradation or (2) ERS-UPR induced apoptosis. Based on the apoptosis of chondrocytes transfected with *NPR2* mutants, it was speculated that *NPR2* gene mutants were likely to promote ERS-UPR-related apoptosis. To verify this hypothesis, *NPR2^WT^*/*NPR2^G602W^*, *NPR2^WT^*/*NPR2^V970F^*, *NPR2^WT^*/*NPR2^R767*^*, *NPR2**^R363*^*/*NPR2^F857S^*, *NPR2^Y306S^*/*NPR2^R557C^*, and *NPR2^vector^* as a negative control were overexpressed in ATDC5 cells. Tunicamycin, a widely recognized N-glycosylation inhibitor, served as an ERS positive control. Expression of glucose-regulated protein, 78 kD (GRP78) and p-IRE1α in chondrocyte-expressing *NPR2* mutants increased significantly, at a level similar to Tunicamycin treated cells (Figure 6C). These results indicated that the *NPR2* mutant activated the ERS-UPR. 

## 4. Discussion 

The *NPR2* gene plays a critical role in human skeleton development, and its mutation can result in different degrees of short stature [19,20]. We found *NPR2* gene pathological variants were diverse and mostly novel, and we found the first evidence that chondrocyte apoptosis and cell differentiation repression through ER stress and the unfolded protein response can be responsible for short stature. 

Loss of function in *NPR2* can result in isolated short stature. Previous studies described that short stature in heterozygous *NPR2* mutations manifested a variable phenotype, considering the differences in short stature severity, body proportion, and some nonspecific skeletal abnormalities, even among members of the same family [21]. The dominant-negative effect is a classical molecular mechanism in genetic diseases, which is defined as the variant that will adversely affect the co-expressed wild-type activity. Patients with heterozygous NPR2 mutations have short stature, also indicating a dominant-negative role of these mutants. Exactly how these mutants exert their dominant-negative effect is not clear. We identified seven *NPR2* gene variants, including one de novo and six novel variants. Among them, the R363* and Y306S were harbored in the binding domain; R557C, G602W, and R767* were harbored in the kinase homology domain; V970F and F857S were harbored in the guanylyl cyclase domain. The variants were mainly missense, followed by nonsense mutation [22]. The penetrance of *NPR2* heterozygous mutation was high, but the clinical presentation of patients was variable, and most of the patients showed short stature without skeletal dysplasia; however, patients with skeletal dysplasia similar to Léri-Weill dyschondrosteosis (LWD) were also observed [23]. In our study, three children with heterozygous mutations (G602W, V970F, and R767*) presented with short stature without typical skeletal deformities (HSDS ranged from −2.25 to −5.60), while two children with compound heterozygous mutations (R363* and F857S; R557C and Y306S) exhibited short stature characterized by shorter limbs and altered thoracic or vertebral endplates (HSDS ranged from −3.10 to −5.35). Three children with *NPR2* heterozygous mutations (G602W, V970F, and R767*) were treated with rhGH, and we found differential yearly growth velocity responses ranging from 6.0 cm to 7.2 cm, and height improved by from +2.18 SD to +2.51 SD over 2.5 and 2.9 years in patient 2 and patient 3, respectively, indicating a positive response to rhGH treatment [24]. Corresponding results from other studies showed that patients before puberty initiation with height improvement of 0.7–2.5 SD after 2–6 years of rhGH treatment [25]. While patient 1, even in puberty, had a poor treatment response with a height improvement of 2.5 cm after one-year of treatment, similar findings were reported in Vasques’s study [26]. To date, data on the efficacy of treatment in *NPR2* gene mutation are limited; further studies with more patients are still needed in the future. 

Recessive inactivating mutations or knockout in *Npr2* gene results in growth insufficiency, skeletal dysplasia in mice and rats associated with small proliferative and hypertrophic chondrocyte areas [27,28]. Robinson et al. [29] showed that *NPR2* could regulate chondrocytes proliferation by antagonizing the fibroblast growth factor receptor 3 signaling by inhibiting the MAPK pathway. To further verify the possibilities of chondrocytes’ decreased differentiation or increased apoptosis that cause narrowed growth plate in *Npr2^−/−^* mice, we constructed seven chondrocyte cell lines with above different mutation alleles and found that all the chondrocytes carrying NPR2^MUT^ exhibited significantly decreased differentiation and increased apoptosis. Apoptosis is a form of programmed cell death that enables a cell to direct its own destruction. This form of cell death could be occurred in normal cells and tissues, particularly for the chondrocytes, since the chondrocytes in cartilage could be affected by various microenvironmental changes, such as hypoxia due to decreased distribution of blood vessels in cartilage [30,31]. In our study, cleave caspase 3, a specific marker of apoptosis and an executioner caspase of apoptosis, was also detected in the wild type and vector control group. Compared with the vector control group, the NPR2^MUT^ group showed an increased expression of cleave caspase 3 under the same culture conditions, indicating the NPR2 mutants could promote apoptosis. Our available data showed that increased apoptosis and decreased differentiation in chondrocytes were among the major causes of growth failure in SNSK and AMDM patients.

Our study also revealed that complete or partially functional defective *NPR2* genes cause incorrect protein trafficking from ER to the cell surface associated with incomplete N-glycosylation. Normally, CNP-NPR2 signaling functions in cGMP production; the heterozygous or compound heterozygous mutants cause loss of CNP-induced cGMP response compared to that of *NPR2^WT^* in our study, and similar decreased responses were also found in Bartels and Hume’s study (less than 60%) [32,33]. Previous observation showed that glycosylation in the ER is required for the catalytic domain to fold into an active conformation [34], and this conclusion could partially explain the decreased cGMP generation in our patients’ mutations since the mutations in our study were distributed in different domains not only restrict to the catalytic domain. Our experiment showed that R557C partially trafficked to plasma membrane had incomplete N-glycosylation, while the remaining mutants that were retained in ER showed no evidence of N-glycosylation the mutant NPR2 proteins were completely or partially retained in ER in our study. Thus, it can be assumed that N-glycosylation was likely proportional to the amount of NPR2 protein expression on the plasma membrane, and loss of N-glycosylation was associated with protein defective trafficking from ER to the cell surface.

NPR2 mutants were restrained in the ER, and the changes in ER morphology, which was consistent with Western blot analysis of IRE1α, indicated ER stress was activated. Previous reports have shown that the majority of LOF *NPR2* mutations were retained in the ER. However, the mechanisms and the outcomes of defective NPR2 trafficking have not been studied yet. In our study, we found that UPR, an adaptive response in which cells attempt to overcome the accumulation of misfolded proteins, was activated, as evidenced by elevated expression of GRP78 and p-IRE1α. The excessive accumulation exceeding UPR processing capacity would result in apoptosis [35]. We found that the activation of UPR robustly promoted chondrocytes apoptosis and altered their metabolism in ATDC5 cells, as shown by the increased expression of cleaved caspase 3 and the decreased expression of differentiation marker *COLII*, *COLXa,* and *BMP4* in ATDC5 cells, the essential components of bone malformation. We thus deduce that the subsequent cascades of ERS-UPR could be the underlying mechanism for *NPR2* gene mutation-induced growth failure. 

## 5. Conclusions

We found one de novo and six novel mutations harbored in the *NPR2* gene in five Chinese families, and the rhGH treatment significantly improved the growth of patients before puberty initiation. Our available data showed that the *NPR2* mutants could inhibit differentiation and promote apoptosis in ATDC5 cells. Defective N-glycosylation and subsequently evoked ERS–UPR signaling might play an important role in NPR2 protein alteration-repressed chondrocyte differentiation.

## Figures and Tables

**Figure 1 cells-11-01265-f001:**
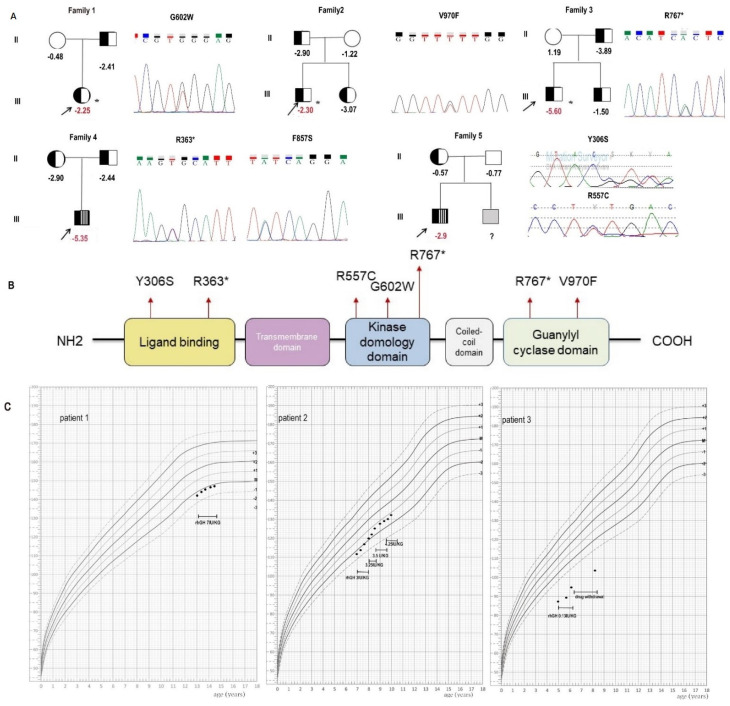
Clinical and genetic findings in five families with short stature. (**A**) Pedigrees of five Chinese families with short stature. (**B**) Diagram showing natriuretic peptide receptor 2 (NPR2) gene structure. The location of the present mutations is represented with a red point. (**C**) Height of the three patients after rhGH treatment. * patients with heterozygous mutations.

**Figure 2 cells-11-01265-f002:**
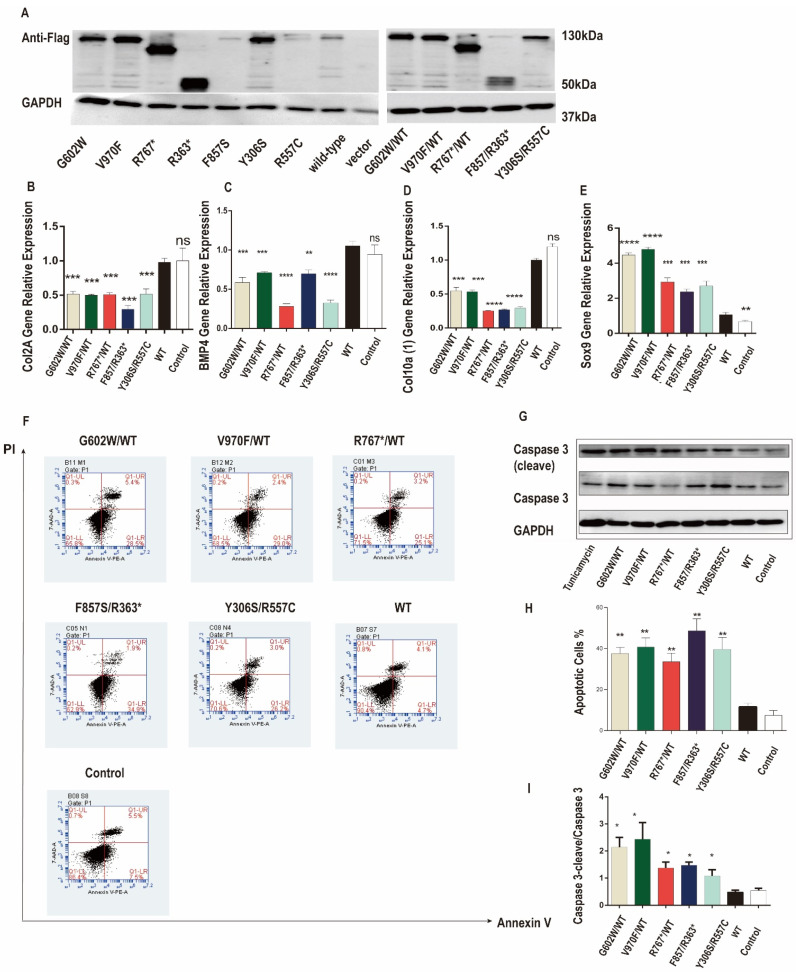
Natriuretic peptide receptor 2 (*NPR2*) mutants inhibit chondrocyte differentiation and promote chondrocyte apoptosis in ATDC5 cells. (**A**) *NPR2^MUT^*, *NPR2^WT^*, and *NPR2^vector^* were transfected in ATDC5 cells for 7 d. The expression of NPR2^MUT/^^WT^ protein is analyzed by Western blot. (**B**–**E**) Real-time polymerase chain reaction (RT-PCR) to determine mRNA levels of collagen type II alpha (*COLIIA*), type X collagen (*COLX*), bone morphogenetic protein 4 (*BPM4*), and SRY-box transcription factor 9 (*SOX9*) in ATDC5 cells 48 h after incubation. (**F**–**I**) Expression of apoptosis-related proteins is detected by Western blot and flow cytometry (Annexin V-FITC/PI). * *p* < 0.05, ** *p* < 0.01; *** *p* < 0.001. and **** *p* < 0.0001. ns—no significance. PI—propidium iodide; GAPDH—glyceraldehyde 3-phosphate dehydrogenase; WT—wild-type; FITC—fluorescein isothiocyanate.

**Figure 3 cells-11-01265-f003:**
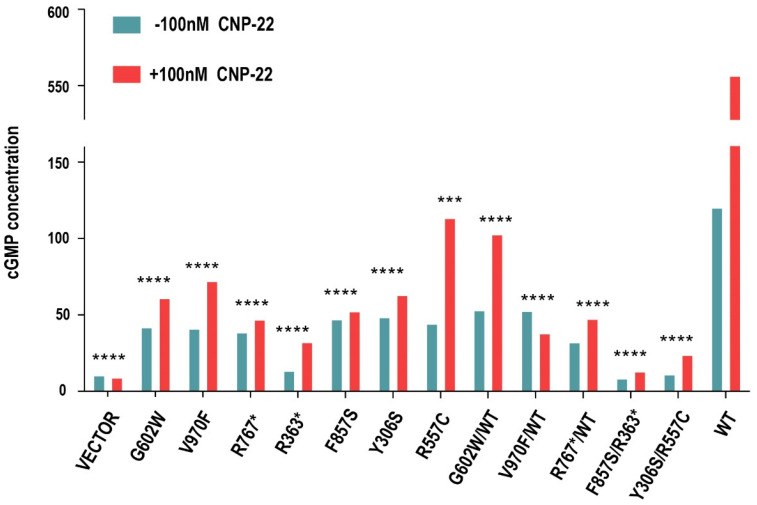
Impaired cyclic guanosine monophosphate (cGMP) production by natriuretic peptide receptor 2 (*NPR2*) mutations. Mutant *NPR2* cells are less effective in producing cGMP. Enzyme-linked immunosorbent assay measuring the guanylate cyclase activity of *NPR2* (wild-type or mutants) expressed in HEK293T. The graph displays average concentrations of cGMP from three independent experiments. *** *p* < 0.001; **** *p* < 0.0001. WT-wild type; GAPDH-glyceraldehyde 3-phosphate dehydrogenase; CNP-22-C natriuretic peptide-22.

**Figure 4 cells-11-01265-f004:**
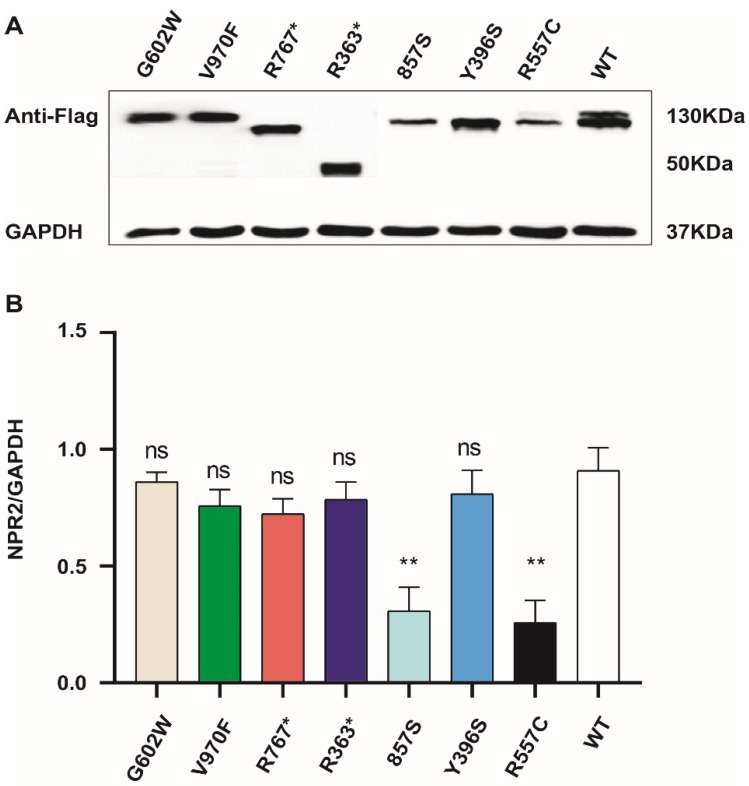
The expression of mutant and wild-type *NPR2* gene in HEK293T cell lines. (**A**) HEK293T cells were transfected with *NPR2^MUT^* and *NPR2^WT^* for 36 h. The expression of NPR2^MUT/WT^ protein is analyzed by Western blot. *NPR2^WT^* and *NPR2^R557C^* typically migrate as two bands in sodium dodecyl sulfate–polyacrylamide gel electrophoresis, while other mutants migrate as only one band. Lanes 3 and 4 indicate the truncated protein. (**B**) Each *NPR2^MUT^* and *NPR2^WT^* band intensity was quantified using Image J software. The graph displays average expression of protein from three independent experiments (wild-type or mutants).** *p* < 0.01; ns—no significance.

**Figure 5 cells-11-01265-f005:**
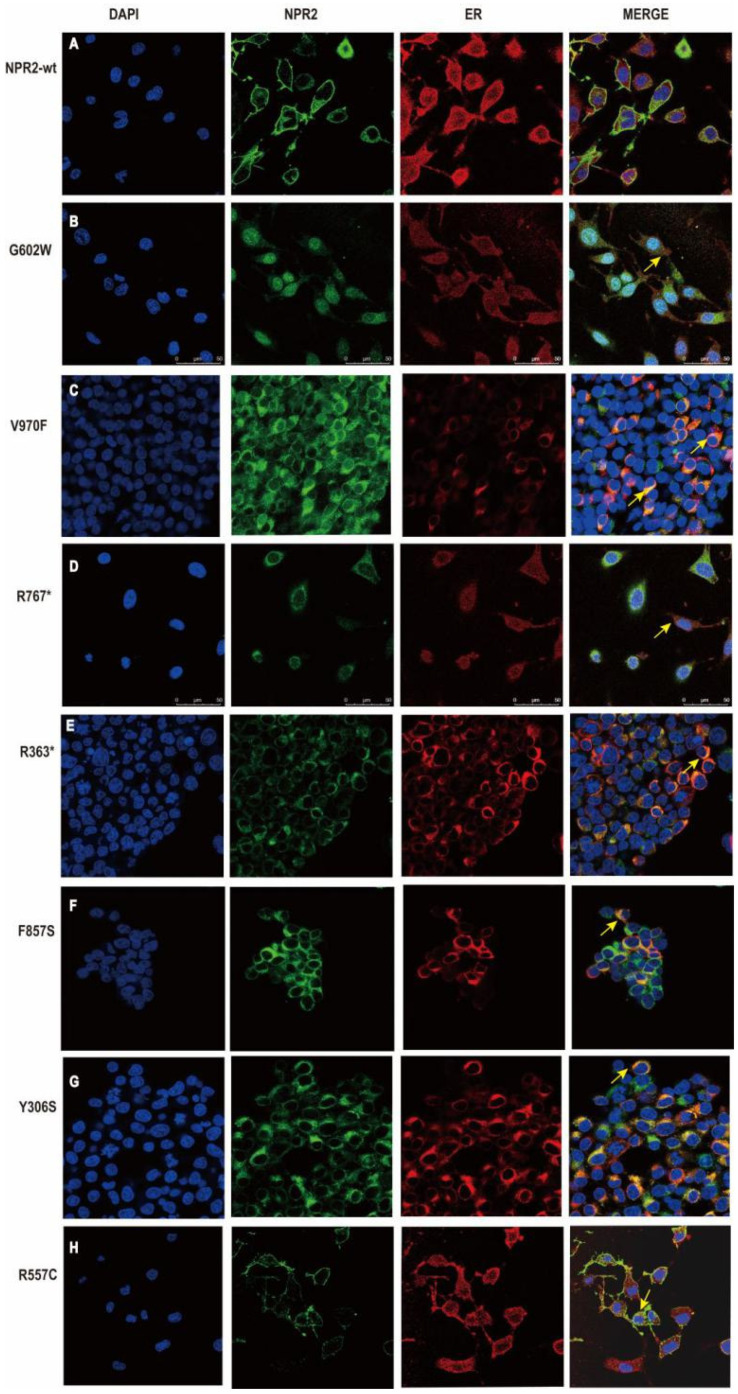
Subcellular localization of natriuretic peptide receptor 2 (NPR2) proteins in HEK293T cells. (**A**–**H**) Except *NPR2^R557C^*, the remaining variants transfected into HEK293T cells lead to accumulation of NPR2 protein (green marker) in the endoplasmic reticulum (ER, red marker). DAPI—diamino-2-phenylindole; WT-wild-type.

**Figure 6 cells-11-01265-f006:**
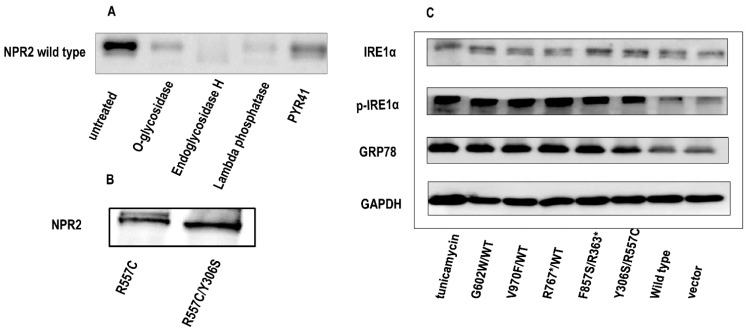
Natriuretic peptide receptor 2 (*NPR2*) mutants induce the endoplasmic reticulum stress-unfolded protein response (ERS-UPR) in ATDC5 chondrocytes. (**A**) ATDC5 cells were transfected with *NPR2^WT^* for 36 h. ATDC5 cell lysates treated with control (no treatment), O-glycosidase, endoglycosidase Endo-H, PYR-41, and Lambda Phosphatase, respectively, and resolved by sodium dodecyl sulfate–polyacrylamide gel electrophoresis are shown. (**B**) N-glycosylation of the R557C *NPR2* mutant. When expressed alone, the R557C mutant is N-glycosylated. However, when co-expressed with the Y306S mutant, the N-glycosylated form of R557C mutant disappears. (**C**) Induction of the UPR by *NPR2* mutants. The expression of glucose-regulated protein, 78 kD (GRP78), and p-inositol-requiring enzyme 1 α (IRE1α) is analyzed by Western blotting in ATDC5 cells transfected with wild-type or mutated *NPR2*. ATDC5 cells incubated with Tunicamycin for 12 h are used as a positive control for ERS and the UPR. WT—wild type; GAPDH—glyceraldehyde 3-phosphate dehydrogenase; PYR-41—a cell-permeable irreversible inhibitor of ubiquitin-activating enzyme E1.

**Table 1 cells-11-01265-t001:** Patient characteristics and their responses to rhGH treatment.

	Proband 1	Proband 2	Proband 3	Proband 4	Proband 5
Gender	Female	Male	Male	Male	Male
Birth weight (kg/SDS)	3.0/−0.10	3.4/+0.26	2.3/−2.63	2.4/+2.37	3.4/+0.26
Birth height (cm/SDS)	48/−1.04	50/−0.24	47/−2.0	51/+0.35	45/−3.81
First visit					
Age (year)	12	7.2	5.2	1.4	2.7
Height (cm)	142.5	113.3	87.0	74.0	73.5
HSDS	−2.25	−2.30	−5.60	−3.1	−5.35
Weight (kg)	38.0	19.4	10.0	10.4	11.0
Tanner stages (before rhGH treatment)	II	I	I	I	I
Peak growth hormone ^c^ (ng/mL)	11.4	14.7	12.0	NA	NA
IGF1(ng/mL)	457	170	72	118	69
IGFBP3 (μg/mL)	5.53	4.38	2.17	3.28	4.34
Bone age (year)	13	7	4	1	2
Phenotypes	No significant dysmorphic features or skeletal abnormities	Radiographs showed no skeletal abnormities	No significant dysmorphic features	Anteroposterior dimension was shorter, changes in many thoracic vertebrae and lumbar vertebrae, short limbs	Changes in the L2 vertebral endplate, short limbs
rGH treatment initial point (year)	13	7	5	/ ^a^	/ ^a^
After rhGH therapy	7 IU/kgd for 1 year	3–4.25/kg/d for 2.5 year	1.25/kg/d for 1.3 year	/ ^a^	/ ^a^
HV (cm/year)	2.5	7.2	6	/ ^b^	/ ^b^
HSDS	−2.92	−1.02	−3.09	/ ^b^	/ ^b^
Tanner stages	II	I	I	/ ^a^	/ ^a^
NPR2 mutation type					
Exon	3	14	8	3/11	1/2
Transcript variant	c.1804G>T	c.2908G>T	c.2299C>T	c.1087C>T c.2570T>C	c.917A>C c.1669C>T
Protein	p. G602W	p. V970F	p. R767*	p. R363* p. F857S	p. Y306S p. R557C
Effect	Missense	Missense	Nonsense	Nonsense Missense	Missense Missense
Novel/known	Novel inherit	Novel inherit	Novel inherit	Novel inherit/Novel inherit	De novo/Novel inherit
Novel/known	Novel inherit	Novel inherit	Novel inherit	Novel inherit/Novel inherit	De novo/Novel inherit

^a^, Parents were unwilling to receive rhGH therapy due to young age; ^b^, no follow-up; ^c^, lower circulating GH levels after the Arginine stimulation test and insulin hypoglycemic test. *, the two early stop mutations in NPR2 (R363* and R767*) lead to truncation of ~280, ~684 amino acids in the C terminus respectively. rhGH—recombinant human growth hormone; HDSD—height standard deviation score; IGF1—insulin growth factor 1; IGFBP3—insulin-like growth factor binding protein 3; HV—height velocity; NPR2—natriuretic peptide receptor 2.

## Data Availability

The data supporting the findings of this study are available upon request from the corresponding author.

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
