# Peer review of "Novel NPR2 Gene Mutations Affect Chondrocytes Function via ER Stress in Short Stature"

_cells, 2022, doi:10.3390/cells11081265_

Round 1
Reviewer 1 Report
General comments:
- In materials and methods, “Cell culture and Transfection”: the TRANSFECTION part is missing. How transfection was performed and how the stable cells were selected are not mentioned. Similarly, the authors have also not mentioned the cell differentiation assay clearly in materials and methods.
- In most of the figures “Times new Roman” font is used making it hard to read. It is recommended to use “Arial” font instead of Times new Roman in Figures.
- Figure 2: different fonts types are used.
- Florescent images in Figure 4 are not of high resolution and therefore, high resolution and readable images should be provided in the online version of accepted manuscript. This is important since the authors are talking about the cellular distribution of NRP2 protein which is not very clear from the current fluorescent images.
Specific comments:
Current study involves the identification of novel NRP2 gene in patients with short stature. The gene is, subsequently, suggested to be involved in ER stress and UPR leading to repression of chondrocyte differentiation. However, some major supporting experiments verifying the above mentioned claims are missing in the current status of the manuscript. For example:
- In Figure 2, with the help of certain cell differentiation markers, authors have shown that NPR2 mutations inhibited chondrocytes differentiation. Later in the same figure, authors have jumped onto the apoptosis assays without giving any connection(s). The more relevant here would be to include either proliferation and/or cell morphological studies.
- Theoretically, cleaved caspase 3 is only visible in apoptotic cells, however, we clearly see an unexpectedly large band in vector transfected cells (Figure 2F). Authors seems to have overlooked it as they did not mention that in results part and not in the discussion.
- It would also be good to quantify active Caspase 3 instead of total caspase as the total caspase may be misleading.
- Importantly, since, major part of the manuscript is based on the claims that NRP2 gene mutations can lead to the apoptosis of the cells, it would be very good if authors could include the data about the activation of other apoptosis markers particularly BAX and Bcl2.
- Caspase 3 activation has been shown to be associated with clear and huge change in DNA morphology including its condensation and fragmentation. However, no such changes have been observed in the Figure 4 (DAPI staining).
- Section “Molecular Characterization of NPR2 gene variants” is more about the impact of NPR2 gene mutants on cGMP level and not about the NPR2 gene characterization. Moreover, Figure 3A, is more relevant to the Figure 4 (see the comments below) or it should be made a separate figure giving the quantification or expression level of each protein including wt and mutants.
- Lentiviruses are used for the transfection of cells which stably transfect cells. Therefore, it is expected that all the cells would be expressing proteins however, many mutant cells for example; R363*, R767* show either no or undetectable expression level of specified proteins.
- Figure 4, Y306S MERGE image and DAPI images are not of the same cells as indicated in NRP2 and ER channels. Hopefully it is a mistake. Please correct it.
- Importantly, Figure 4 based conclusion on accumulation of NRP2 in ER is not enough. There is apparently no colocalization between ER marker and NRP2. This applies to both wt and mutant proteins. If the wt protein is processed through ER we should expect some colocalization with ER also and the mutant should be more in colocalization, however, in these figures no colocalization can be observed.
- ER stress has been shown to be associated with morphological impact on ER and mitochondria, which is apparently clear in the Figure 4, however, no reference is made to that and it has not been discussed in the discussion section.
Author Response
We thank the reviewer for your constructive remarks and suggestions, which, in our opinion, has significantly improved the quality of the manuscript. Please see the attachment.

Reviewer 2 Report
Cells (ISSN 2073-4409)
Manuscript ID cells-1631981
Novel NPR2 gene mutations identified in patients with short stature repress chondrocyte differentiation by inducing ER stress and the unfolded protein response
Feihong Luo * , Qiuyue li , Xin Fan , Wei lu , Chengjun sun , Zhou Pei , Miaoying Zhang , Jinwen ni , Jing Wu , Faxing yu *
The present study focuses on examination the function of Novel NPR2 gene mutations that lead to acromesomelic dysplasia, Maroteaux type (AMDM), short stature with nonspecific skeletal abnormalities (SNSK), and epiphyseal chondrodysplasia, Miura type (ECDM). Authors identified one de novo (R557C) and six novel variants (G602W; V970F; R767*; R363*; F857S; and 19 Y306S) in five independent Chinese families with familial short stature. Authors found that ATDC5 cells expressing mutant NPR2 gene showed decreased differentiation and elevated apoptosis rate in ATDC5 cells. They further showed that the aberrant N-glycosylation modification, plasma membrane localization, and ER stress resulted from the accumulation of NPR2 mutant protein in ER. The authors claim that NPR2 loss of function induces chondrocyte apoptosis and represses cell differentiation through ER stress and the unfolded protein response.
There are several concerns to be addressed as listed.
Major concerns:
- Figure 1, authors claim that R557C as a de novo The NPR2 genotypes of parents of the proband should be provided.
- Figure 2, The expression levels of NPR2 mutants in ATDC5 cells should be provided before the functional assays, as well as the endogenous expression levels of NPR2 in ATDC5 (the vector-transfected only controls should be also included).
- The NPR2 mutants inhibit chondrocyte differentiation and induce chondrocyte apoptosis were observed in ATDC5 cells. Why the molecular characterization was done in HEK239T cells instead of ATDC5 cells? Do NPR2 mutants also reduce the CNP-induced cGMP response compared to that of NPR2WT.
- The NPR2MUT affected N-glycosylation, induced UPR and ER stress in ATDC5 cells that might also induce protein degradation of NPR2MUT. Furthermore, protein abundance greatly contributes to protein function. This reviewer noticed that NPR2 WT protein treated with N-glycosylation inhibitors significantly reduced the NPR2 WT protein accumulation and inhibition of ubiquitination increased protein accumulation in ATDC5 cells expressing NPR2 (Figure 5A). Whether protein abundance of NPR2MUT was affected in ATDC5 cells?
- Several NPR2 loss-of-function mutations identified previously that
- The authors claim that NPR2 loss of function induces chondrocyte apoptosis and represses cell differentiation through ER stress and the unfolded protein response. However, three children with heterozygous mutations have short stature, and cells expressing different NPR2 mutants and WT NPR2 (e.g. G602W/WT) increased ER stress, affected the chondrocyte differentiation and apoptosis. A dominant-negative effect that causes a short stature phenotype in the heterozygous state in AMDM families should be discussed.
Minor issues:
- Line 76, spell out WES.
Line 86, spell out ES.
Author Response

(The authors gave the same response as above.)

Reviewer 3 Report
This is a translational study with clinical-genetic and cell-culture investigations regarding mutations in NPR2 and NPR2 function in vitro.
My review will focus on the clinical-genetic aspects of the manuscript.
The abstract and introduction are very well written. The materials and discussion could benefit from a language edit.
My main concern regarding the clinical presentation of the work is that the patient data is not sufficiently described. Growth is a function over time and it would be very helpful if authors could show the growth curves of the 3 older children who were treated with rGH. Also, it would be helpful to give information on pubertal status, parental heights, possible changes in dosing of rGH, development of scolisosis during treatment etc.
It is unclear how these 5 families were chosen/identified. Is every child/family in endocrine with short stature undergoing WES and the 5 with the NPR2 mutations were further investigated?
While in the first part the clinical data and response to rGH is presented, there is almost no discussion of these observations in the discussion/conclusion. It might not be feasible to connect the clinical and the cell-culture basic science part of this manuscript. And that is ok from my point of view. However, both parts need to be able to ‘stand alone’ – and the clinical part needs some improvement as outlined above. However, perhaps authors also want to hypothesize/discuss the meaning of their findings on a cellular level for a potential response to rGH on a clinical level.
Specific comments:
Materials and Methods:
- Pedigree analysis: WES is not introduced as an abbreviation. Is whole exome sequencing meant?
- Genetic testing: ES is not introduced, but possibly WES was meant?
Table 1:
- Please provide SDS for Birth measures and weight.
- Does Peak Growth hormone refer to a growth hormone stimulation test? If so which one.
- When was rGH treatment commenced? Is it possible that patient 1 was too old to benefit from GH treatment?
- Is the growth velocity calculated for the entire period of rGH treatment? Or only the last or first year?
Figure 1: there are no clinical findings in this figure displayed. Pls change title.
Figure 2:
- the colors of the bars seem to differ between the different panels in the first row. Pls adjust.
- What is the difference between wildtype and control? Empty vector?
- Can authors use the same color coding for the same cell lines throughout the figure?
Discussion:
- Authors are discussing a potential effect of advanced pubertal development on the effect of growth hormone treatment. This is a relevant aspect and pubertal status of the treated children in this study at beginning and end of the GH therapy should be provided.
Supplementary Figure 1: This is a very informative figure. Please provide more information e.g. on the age of the patient at the time of X-ray.
Author Response

(The authors gave the same response as above.)

Round 2
Reviewer 1 Report
The manuscript is now improved
Author Response
So many thanks for your affirmation and approval to our article. Thank you very much again for your great efforts on our manuscript.
Reviewer 2 Report
The quality of the manuscript has significantly improved.
Authors described that “The NPR2 genotypes of parents of the proband has been added as the supplementary Figure 3 in supporting information.”
However, supplementary Figure 3 in supporting information was missing in the revised manuscript.
Author Response
Thank you very much for your kind review. The Supplementary Figure 3 has been provided, including the information of NPR2 (de novo R557C mutation) genotypes of parents of the proband. Please see the attachment.
